# Laser flash melting cryo-EM samples to overcome preferred orientation

Monique S. Straub [1,2], Oliver F. Harder[1,2], Nathan J. Mowry[1,2], Sarah V. Barrass [1], Jakub Hruby [1], Marcel Drabbels [1] & Ulrich J. Lorenz [1] ✉

Sample preparation remains a bottleneck for protein structure determination by cryo-electron microscopy. A frequently encountered issue is that proteins adsorb to the air–water interface of the sample in a limited number of orientations. This makes it challenging to obtain high-resolution reconstructions, or may even cause projects to fail altogether. We have previously observed that laser flash melting and revitrification of cryo-EM samples reduces preferred orientation for large, symmetric particles. Here we demonstrate that our method can in fact be used to scramble the orientation of proteins of a range of sizes and symmetries. The effect can be enhanced for some proteins by increasing the heating rate during flash melting or by depositing amorphous ice onto the sample prior to revitrification. This also allows us to shed light onto the underlying mechanism. Our experiments establish a set of tools for overcoming preferred orientation that can be easily integrated into existing workflows.

Since the resolution revolution, cryo-electron microscopy (cryo-EM) has greatly expedited the process of protein structure determination[1,2]. Advances in instrumentation and single-particle analysis now routinely enable high-resolution reconstructions for a wide range of proteins and protein complexes[3–6]. However, sample preparation issues have remained a major bottleneck[7]. In particular, one common challenge involves preferred particle orientation[8–10]. When the sample solution is applied to the cryo-EM specimen support, the hydrophobic regions of the protein surface adsorb to the air–water interface, so that only a limited number of viewing directions are present after vitrification. Strong preferred orientation limits the resolution of a single-particle reconstruction along certain viewing directions and can even make it difficult to obtain a reconstruction at all[11].

Although several strategies can address preferred orientation issues, no universal solution has been found. Tilting the sample provides a wider range of particle views, but usually results in larger beam-induced motion, an increased ice thickness in the viewing direction and a defocus gradient across the viewing area[12,13]. Preferred orientation can be reduced by covering the air–water interface with detergents or small proteins to block adsorption of the particles of interest[14,15]. Moreover, different specimen supports, such as graphene,

graphene oxide or other functionalized surfaces, can prevent some particles from diffusing to the interface[16–20]. However, adopting such strategies often requires time-consuming reoptimization of the sample preparation procedure. Adsorption at the air–water interface can also be minimized by limiting the time between sample application onto the specimen support and vitrification[21–26]. Although this interval can be shortened to a few milliseconds, particles generally diffuse to the interface within a few microseconds. Substantially faster preparation speeds would therefore be required to prevent the majority of particles from reaching the interface. Finally, machine-learning techniques have been developed to guess missing views[27–29]. However, such approaches cannot replace actual experimental observations.

We have previously observed that flash melting and revitrification of cryo-EM samples with microsecond laser pulses reduces preferred orientation for some large, highly symmetric particles[30–32]. Flash melting seems to exert small forces on the proteins, altering their orientation and resulting in an uneven spatial distribution after revitrification, with particles clustering together in some areas[30,32]. High-resolution reconstructions suggest that the revitrification process preserves particle integrity and does not impose a fundamental limit on the obtainable spatial resolution[30,33]. However, the practicality of using

[1]Ecole Polytechnique Fédérale de Lausanne (EPFL), Laboratory of Molecular Nanodynamics, Lausanne, Switzerland. [2]These authors contributed equally: Monique S. Straub, Oliver F. Harder, Nathan J. Mowry. ✉e-mail: ulrich.lorenz@epfl.ch

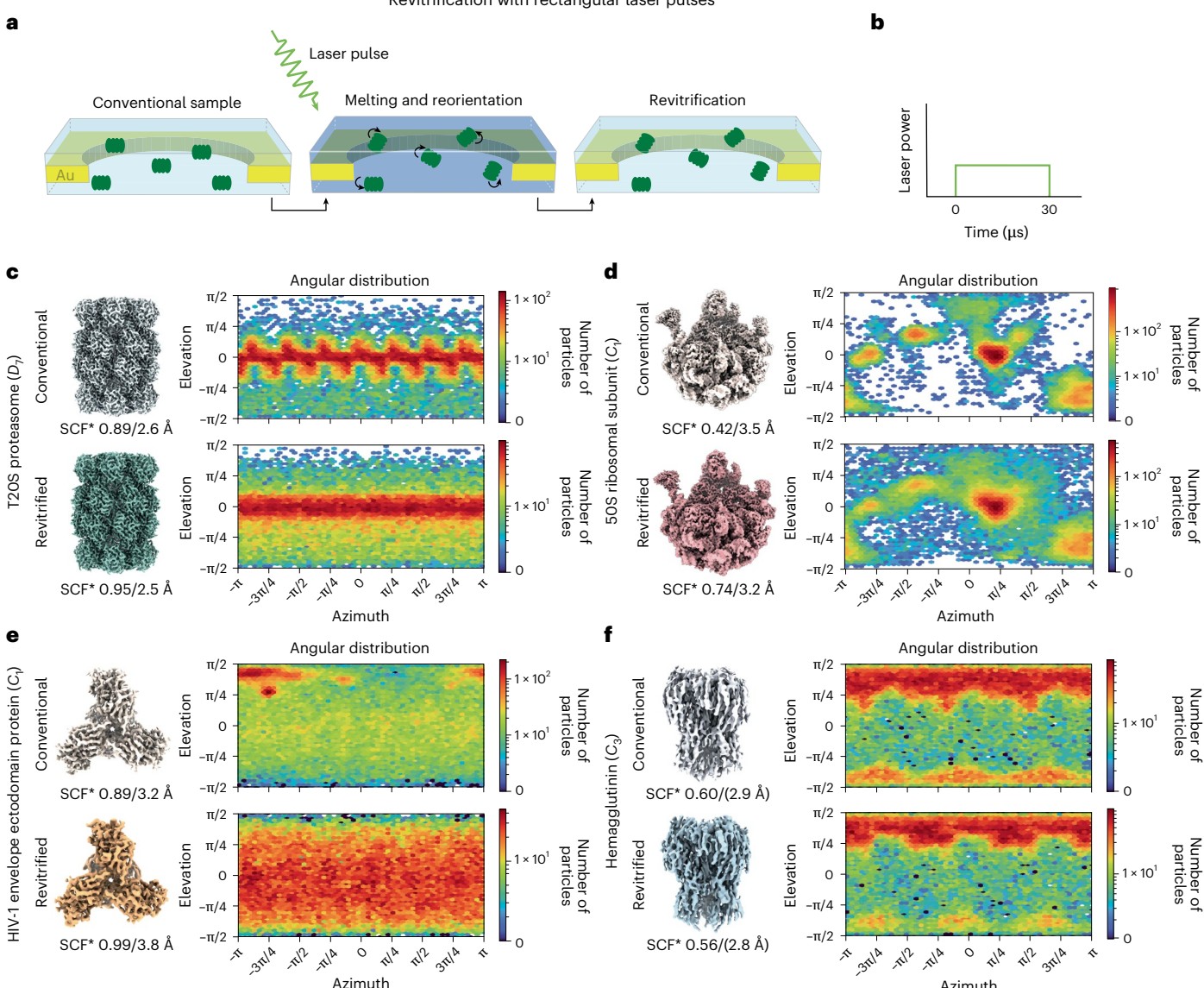

**Fig. 1 | Revitrification with rectangular laser pulses. a**, A cryo-EM sample is flash melted with a 30-μs laser pulse, which scrambles the orientation of the particles. When the laser is switched off, the sample revitrifies, trapping the particles in a non-equilibrium angular distribution. **b**, Schematic of the rectangular laser pulse shape. **c**–**f**, Reconstructions, with the applied symmetry noted in parentheses, and angular distributions of conventional and revitrified samples (top and bottom, respectively) for the T20S proteasome (**c**), 50S ribosomal subunit (**d**), HIV-1 envelope ectodomain protein (**e**) and hemagglutinin (**f**). The SCF*, which is a measure of the degree of preferred orientation, as well as the resolution of the reconstructions are indicated. For hemagglutinin, the resolution appears to be overestimated, as judged by the quality of the map.

flash melting to address preferred orientation and the underlying mechanism remain unclear. Here, we show that this effect extends to smaller proteins, as well as particles with lower symmetry. We also introduce two variants of the experiment that provide larger changes in preferred orientation for some proteins and that allow us to elucidate the underlying mechanism.

## Results

### Flash melting with rectangular laser pulses

Our experimental approach is illustrated in Figure 1a (refs. 30,32,34,35). A cryo-EM sample is flash melted with 20-μs or 30-μs rectangular laser pulses, where the laser power is kept constant for the pulse duration (Fig. 1b). This causes some particles to detach from the air–water interface and change their orientation. Once the laser beam is switched off, the sample cools in just a few microseconds and revitrifies[35,36], trapping the proteins in a non-equilibrium angular distribution (Fig. 1a).

We performed in situ flash-melting experiments following previously described protocols, using a transmission electron microscope that we have modified for time-resolved experiments[37,38]. The sample is locally flash melted by aiming the laser beam (532 nm wavelength, 28-μm diameter spot size in the sample plane) at the center of a grid square of the holey gold specimen support (1.2 μm holes, 1.3 μm apart on 300-mesh gold). Typically, an area of 9–16 grid holes is revitrified. The samples are then transferred to a high-resolution electron microscope for imaging (Supplementary Figs. 1, 2, 4, 5 and 9–12 and Supplementary Table 1).

Figure 1c demonstrates that flash melting reduces preferred orientation for a sample of the T20S proteasome. A reconstruction from a conventional sample is shown at the top, together with the angular distribution of the particles. It exhibits a series of maxima that arise from the T20S proteasome's strong preference to adsorb to the air–water interface on its side, resulting in sparse population of particles in the top views. Upon revitrification (bottom), the angular distribution

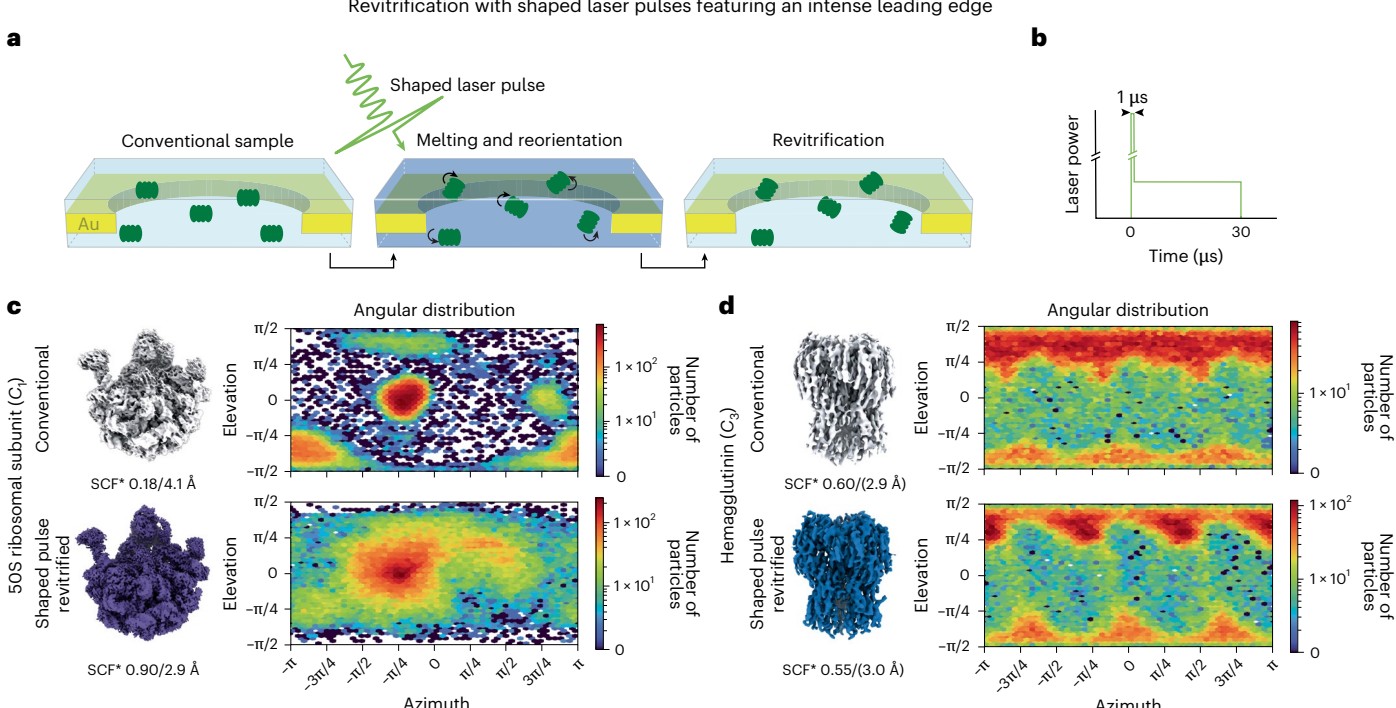

**Fig. 2 | Revitrification with shaped laser pulses featuring an intense leading edge. a**, A cryo-EM sample is flash melted with a shaped 30-μs laser pulse that features an intense leading edge. When the laser is switched off, the sample revitrifies, and the angular distribution of the particles is reshuffled. **b**, Schematic of the laser pulse shape, which features an intense spike at its onset (1-μs duration with the laser power increased ten-fold). **c**,**d**, Reconstructions and angular distributions of conventional and revitrified samples (top and bottom, respectively) for the 50S ribosomal subunit (**c**) and hemagglutinin (**d**). The SCF* and the resolution of the reconstructions are shown below the maps. For hemagglutinin, the resolution appears to be overestimated, judging by the quality of the map.

broadens noticeably, with a larger fraction of the particles populating the less-favored views. At the same time, the sampling compensation factor (SCF*), a measure of the degree of preferred orientation, improves from 0.89 to 0.95. The SCF* takes values between 0 and 1, with 1 corresponding to a perfectly isotropic angular distribution[5,39,40].

An even greater reduction in preferred orientation can be achieved for the 50S ribosomal subunit. Unlike the highly symmetric T20S proteasome (point group $D_7$), the asymmetric 50S ribosomal subunit ($C_1$) exhibits strong preferred orientation (Fig. 1d). In a conventional sample, the angular distribution features a small number of pronounced maxima, with many views absent altogether (white) and a low SCF* of 0.42. Upon revitrification, most of the missing views become populated, and the SCF* improves to 0.74. At the same time, the streaky artefacts that are visible in the reconstruction from the conventional sample and that are the result of preferred orientation disappear (Supplementary Fig. 5)[41]. Flash melting can also improve the angular distribution of much smaller proteins, such as the HIV-1 envelope ectodomain protein (210 kDa), for which the SCF* increases from 0.89 to 0.99 (Fig. 1e). Among these examples, a noticeable gain in resolution is observed only for the 50S ribosomal subunit, the only protein for which the preferred orientation is so pronounced that many views are not populated at all. For the HIV-1 envelope ectodomain protein, the resolution slightly decreases. This seems to be largely due to astigmatism in the micrographs recorded for the revitrified sample.

Laser flash melting seems to be less efficient at reducing preferred orientation for proteins that are more strongly bound to the air–water interface. Hemagglutinin (170 kDa) exhibits a preference for top and bottom views (Fig. 1f). Although revitrification depopulates particles in the bottom views, the overall angular distribution remains similar, with the SCF* slightly decreasing from 0.60 to 0.56. This suggests either that flash melting detaches only a few proteins from the interface or

that detached particles return to their preferred orientation rapidly after diffusing to the surface. Both explanations indicate that there is a strong interaction between hemagglutinin and the air–water interface. For the proteins studied here, diffusion to the interface is expected to occur in just a few microseconds, which is shorter than the duration of the experiment (30 μs).

**Flash melting with shaped laser pulses**

Experiments with shaped microsecond laser pulses provide clues about the mechanism through which the angular distribution of the particles is reshuffled. We have previously shown that cryo-EM samples partially crystallize during flash melting with rectangular laser pulses[35,42]. This led us to speculate whether the transient formation of crystallites, which preferentially form at the sample surface[43], might exert small forces on the particles, leading to their detachment and reorientation[30]. We test this hypothesis by melting samples with shaped laser pulses that feature an intense 1-μs initial spike and ten times the laser power (Fig. 2a,b)[44]. This increases the heating rate during flash melting to about $2 \times 10^8$ K s$^{-1}$, twice the critical heating rate of $1 \times 10^8$ K s$^{-1}$, so that the sample does not crystallize[44]. If crystallization was the dominant mechanism for reshuffling the particle orientations, we would expect the angular distribution to barely change under these conditions. Contrary to our expectation, flash melting with such a shaped laser pulse decreases preferred orientation even more drastically for the 50S ribosomal subunit (Fig. 2c and Supplementary Figs. 7 and 8). The SCF* increases from 0.18 to 0.90 in revitrified areas of the sample, and the map resolution improves markedly, from 4.1 Å to 2.9 Å. Of note, the angular distribution of the sample is different from that in Figure 1d, despite both being prepared using the same procedure. This suggests that the interaction of the particles with the interface is very sensitive to preparation conditions and can be easily altered.

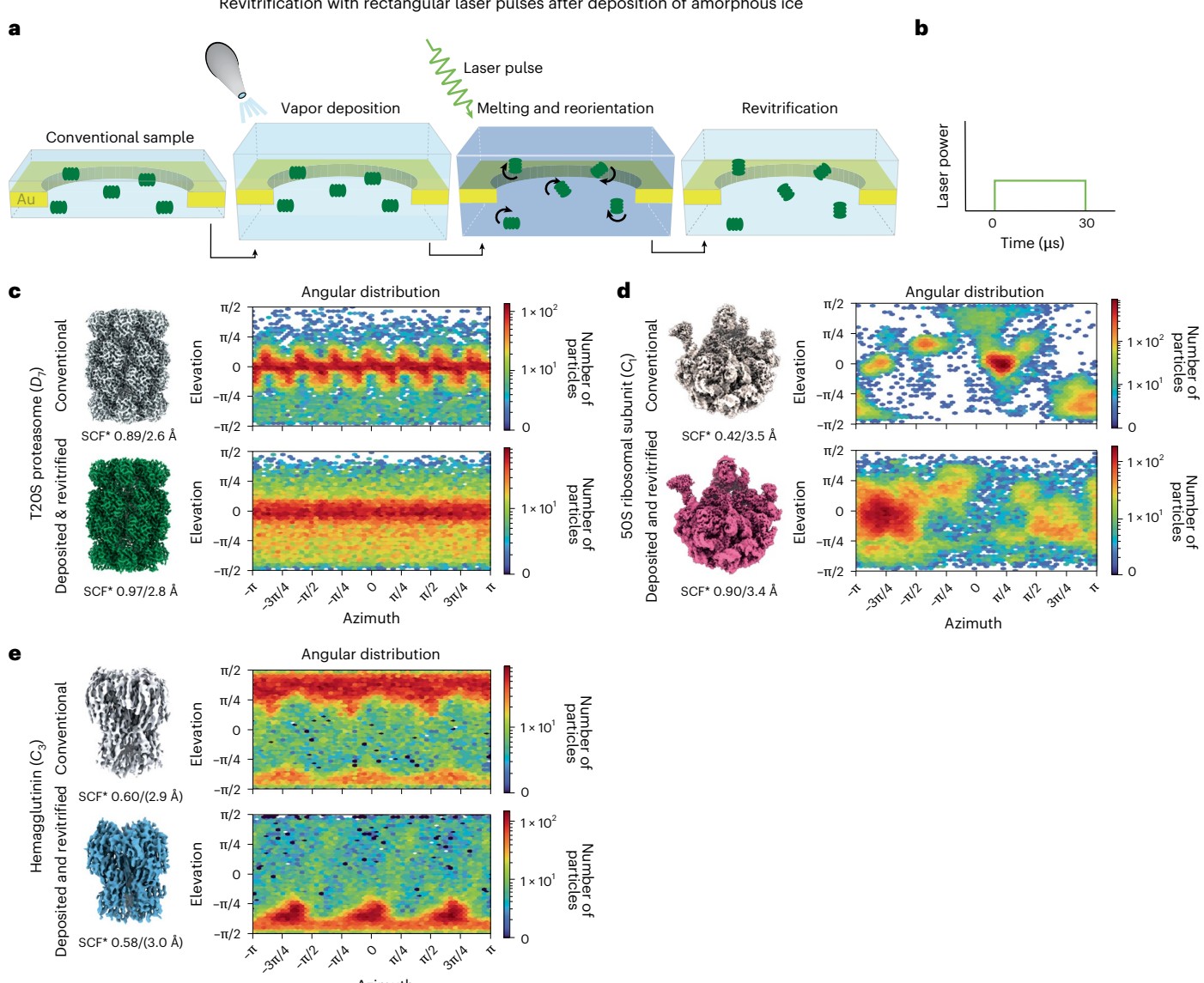

**Fig. 3 | Revitrification with rectangular laser pulses after deposition of amorphous ice. a**, A 20-nm layer of amorphous ice is deposited in situ onto the cryo-EM sample before it is flash melted with a 30-μs laser pulse. When the laser is switched off, the sample revitrifies with the particle orientations redistributed. **b**, Schematic of the rectangular laser pulse shape used. **c**–**e**, Reconstructions and angular distributions of conventional and revitrified samples (top and bottom, respectively) for the T20S proteasome (**c**), the 50S ribosomal subunit (**d**) and hemagglutinin (**e**). The sampling compensation factor (SCF*) and the resolution of the reconstructions are indicated. For hemagglutinin, the resolution appears to be overestimated, judging by the quality of the map.

Clearly, transient crystallization of the sample cannot be the dominant mechanism for particle reorientation. Instead, the effect appears to intensify with increased heating rates, suggesting several possible mechanisms. Impulsive laser heating of a thin, suspended membrane, such as the holey gold film of the specimen support, induces drumming motions that typically persist on microsecond timescales[45,46]. With the larger initial heating power of the shaped laser pulses, such oscillations likely reach a larger amplitude and exert greater forces on the thin liquid film in which the particles are suspended. This could allow them to detach from the air–water interface more readily and reorient. The impinging laser beam also deforms the thin liquid film directly through the radiation forces[47]. Additionally, it is also conceivable that some particles detach from the interface owing to the evaporation of the liquid sample in the vacuum of the electron microscope[42]. Simulations indicate that, during the initial spike of the shaped laser pulse, the sample temperature briefly increases by about 25 K (Supplementary Fig. 15). Because the evaporation rate increases

exponentially with temperature[48], this could cause more particles to detach. However, there are also changes in preferred orientation when samples are revitrified at atmospheric pressure, under which evaporation is substantially reduced[33,49]. It is therefore unlikely that evaporation is the dominant effect that causes particles to detach from the air–water interface. For hemagglutinin, even shaped laser pulses cannot reduce the amount of preferred orientation, although they induce some changes in the angular distribution (Fig. 2d and Supplementary Figs. 11 and 14).

## Flash melting after deposition of a layer of amorphous ice
Our experiments suggest that the changes in the angular distribution induced by flash melting are the result of two opposing processes. First, particles must be successfully detached from the air–water interface to reorient. But given enough time, they can then diffuse back to the interface and return to their preferred orientation. A simple experiment allowed us to detach all particles from the interface before laser melting

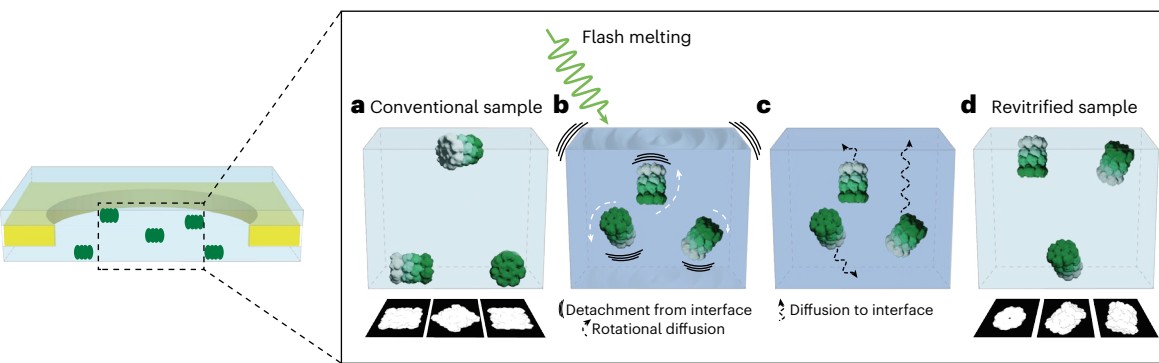

**Fig. 4 | Competing processes determine the degree to which laser flash melting is able to scramble the particle orientations. a**, In a conventional cryo-EM sample, particles adsorb to the air–water interface in preferred orientations. **b**, Flash melting detaches the proteins from the interface, so that they are able to rotate freely. **c**, While the sample is liquid, many particles diffuse back to the interface, where some adsorb in their preferred orientation. **d**, The sample is revitrified and the ensemble is arrested in a non-equilibrium angular distribution.

the sample, disentangling both processes (Fig. 3a). We deposited a 20-nm-thick layer of amorphous ice onto both sides of the sample by introducing water vapor into the volume surrounding the specimen in our modified transmission electron microscope[50]. After flash melting the sample with a rectangular laser pulse (Fig. 3b), the particles all began at a distance from the interface, having lost all memory of where they used to be located. The angular distribution obtained after revitrification then exclusively reflects the result of the particles' free diffusion.

When we revitrify a T20S proteasome sample after depositing a layer of amorphous ice, we obtain the angular distribution in Fig. 3c (bottom, Supplementary Fig. 3). The estimated rotational timescale of the particles (about 5 µs) is shorter than the timescale of the experiment, so one might expect that this should yield an isotropic distribution. Instead, the distribution closely resembles that obtained without prior deposition of amorphous ice (Fig. 1c) and notably features a preference for side views. This clearly indicates that the particles have diffused back to the interface, which occurs on an estimated timescale of only about 4 µs, and have partially settled into their preferred orientations. The striking similarity of the angular distribution to that in Figure 1c suggests that both are the result of the free diffusion and partial readsorption of the particles. It therefore appears likely that, even without prior deposition of amorphous ice, the laser pulse efficiently detaches the T20S proteasome particles from the interface.

Flash melting of a cryo-EM sample of the 50S ribosomal subunit, following the deposition of amorphous ice, similarly results in a broad angular distribution with a small degree of preferred orientation, indicating that some particles have readsorbed to the interface (Fig. 3d and Supplementary Fig. 6). The SCF* increased from 0.42 to 0.90. Curiously, however, the distribution substantially changed, featuring maxima at orientations that were previously only sparsely populated. This suggests that the interaction between the particles and the interface was altered in our experiments. As discussed above, the preferential orientation of the 50S ribosomal subunit varies, even in conventional samples prepared under otherwise identical conditions (Figs. 1d and 2c), suggesting that the properties of the interface are easily modified. Sample transfer and handling could have introduced traces of surface-active compounds, thereby modifying the chemical composition of the interface. Additionally, the interaction between the particles and the surface could have been affected by the dilution of the buffer during our experiment. At the laser powers used, less material is evaporated than has been deposited, so that the sample thickness slightly increases overall. As shown in Figure 3e, we obtain a similar result for hemagglutinin (Supplementary Fig. 13). After amorphous ice deposition and revitrification, the particles show a substantial amount of preferred orientation, indicating that many have readsorbed to the interface. However, the prominent top views have been largely depopulated, whereas the bottom views now dominate the angular distribution.

## Discussion

Our experiments yield a simple picture of the competing processes that occur during laser revitrification. Flash melting detaches some particles from the interface (Fig. 4a,b), a process that becomes more efficient as the heating rate is increased. As these particles rotate freely, their orientations are scrambled, resulting in improved angular distribution. In a competing process, particles diffuse back to the interface (Fig. 4c) before the sample is revitrified (Fig. 4d), allowing some proteins to efficiently readsorb in their preferred orientation. This process generally is more efficient for small proteins because of their shorter diffusion times. It could be possible to improve the angular distribution of such proteins by reducing the duration of the laser pulse to below the diffusion timescale (a few microseconds), so that the particles do not have enough time to reach the interface. In some experiments, proteins readsorbed in a different preferred orientation, such as when we deposited a layer of amorphous ice prior to revitrification. Apparently, our experiment has altered the interfacial properties, making adsorption in new orientations favorable. This implies that it should be possible to purposely modify the interface before flash melting, for example through vapor deposition of a hydrophilic compound onto the cryo-EM sample. Upon laser melting, all particles should then desorb and randomize their orientation. Such experiments are currently underway in our lab[51].

Our experiments also provide a practical toolbox for addressing preferred orientation, with some proteins showing dramatic improvements in their angular distribution. For example, revitrifying a cryo-EM sample of the 50S ribosomal subunit with shaped laser pulses improves the resolution by 1.2 Å under otherwise identical conditions (Fig. 2c). To put this in perspective, 18 times as many micrographs of a conventional sample would be needed to achieve the same improvement. Reshuffling the particle orientations through laser flash melting can therefore provide considerable cost and time savings. Our method can be easily combined with other established approaches for addressing preferred orientation. Importantly, as a simple physical approach, flash melting does not require any time-consuming changes to the sample-preparation procedure, but can be easily integrated into existing workflows. As we have previously shown, cryo-EM samples can also be revitrified in an optical microscope equipped for correlative light and electron microscopy experiments[33]. This approach is technically less involved and might therefore be more accessible to other labs. It could also be integrated into vitrification devices, some of which already use an optical microscope to assess sample quality. Laser-melting experiments could be directly integrated into

high-resolution electron microscopes at cryo-EM facilities. If a sample is found to exhibit preferred orientation during data acquisition, it could then be revitrified on the fly with little additional time.

## Online content

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

## Methods

### Cryo-EM sample preparation

Purified T20S proteasome (3.7 mg ml$^{-1}$ in 50 mM Tris pH 8.0, 200 mM sodium chloride) and purified 50S ribosome[52] (an optical density at 260 nm (OD$_{260}$) of 40 per ml in 20 mM HEPES pH 7.5, 100 mM sodium chloride, 2 mM magnesium chloride) were gifts from A. Guskov, University of Gröningnen, and B. Beckert, DCI Lausanne respectively. The purified hemagglutinin sample (1 mg ml$^{-1}$ in PBS pH 7.5) was purchased from MyBioSource (MBS434205). The cryo-EM samples of purified proteins were prepared by applying 3.5 µl of the sample solution to glow-discharged (0.25 mbar, 15 mA, 60 s) gold grids (UltrAuFoil, R1.2/1.3, 300 mesh, Quantifoil Micro Tools GmbH). Surplus sample was removed by blotting for 2–4 s using a blot force of 10 in a Vitrobot Mark IV (Thermo Fisher Scientific, held at 20 °C, 100% relative humidity). The HIV-1 envelope ectodomain protein (HIV-1 CH505TF Envelope SOSIP ectodomain)[53] was generously provided by R. Henderson, Duke Center for HIV Structural Biology. The HIV-1 envelope ectodomain protein (5 mg ml$^{-1}$ in 15 mM HEPES pH 7.4, 150 mM NaCl, 2.5% glycerol) was applied to glow discharged gold grids (Au-flat, 0.6 µm/1 µm hole diameter/pitch, 300 mesh, Electron Microscopy Sciences). The sample was blotted with the minimum blot time that can be set (0 s). The conventional sample was blotted using a blot force of 10, whereas the revitrified sample was blotted using blot forces of 5 and 10. All samples were plunge-frozen in liquid ethane and stored in liquid nitrogen until experiments were carried out.

### Flash melting with rectangular laser pulses

The revitrification experiments with rectangular laser pulses (Fig. 1) were performed in situ using a modified JEOL 2200FS transmission electron microscope equipped with a TemCam-XF416 (TVIPS) and an in-column Omega energy filter[38]. The samples were loaded into the microscope using a single-tilt side-entry cryo-transfer holder (Elsa, Gatan) held at a temperature of approximately 100 K. The sample was melted and revitrified in the center of multiple grid squares (typically 80 – 100) through irradiation with microsecond laser pulses[30,32,35]. A 30-µs laser pulse was used for the 50S and HIV-1 samples, and a 20-µs laser pulse for the T20S sample. The hemagglutinin sample was revitrified with two 30-µs pulses with a 250-ms delay between the pulses. The laser pulses were generated by chopping the continuous output of a 532 nm laser (Ventus532, Laser Quantum) with an acousto-optic modulator (AA Opto-Electronic). The laser beam was focused to a spot size of 28 µm FWHM (full width at half maximum) in the sample plane, as determined by a knife-edge scan, and aligned to the center of the grid squares. The laser power was adjusted on the fly to keep the size of the revitrified region, typically 15–22 µm in diameter, approximately constant[35]. After revitrification, the grids were unloaded from the side entry holder and stored in liquid nitrogen until data collection.

### Flash melting with shaped laser pulses

The revitrification experiments with shaped laser pulses (Fig. 2) were performed on a modified JEOL 2010F transmission electron microscope, equipped with a TemCam-XF416 (TVIPS)[50,54,55]. The plunge-frozen cryo-EM samples were inserted into the microscope using a single-tilt side-entry cryo-transfer holder (Elsa, Gatan) held at a temperature of approximately 100 K. Shaped laser pulses with an intense leading edge (1 µs initial spike at ten times the laser power) were generated by modulating the output of a continuous green laser (532 nm, Verdi Coherent) with an acousto-optic modulator (AA Opto-Electronic) that was controlled by an arbitrary function generator (Tektronix AFG1062)[44]. The laser beam was focused to a spot of 38 µm FWHM in the sample plane, as determined by imaging the beam with a CCD camera placed in a plane conjugate to the sample plane. The laser power required for laser melting was determined by using ordinary rectangular pulses. The initial spike was then added to perform the revitrification experiment.

### Flash melting with rectangular laser pulses after ice deposition

The experiments in Figure 3, in which the sample was revitrified after deposition of amorphous ice, were performed on a modified JEOL 2010F transmission electron microscope, equipped with a TemCam-XF416 (TVIPS)[42,50,54,55]. Amorphous ice was deposited by leaking water vapor into the volume surrounding the sample through a gas dosing valve[50]. The vapor enters the microscope through a stainless steel tube that terminates inside of the cold shield, in close proximity to the sample. To obtain a well-defined thickness, the deposition rate was first determined with the following procedure. A multilayer graphene sample on a gold specimen grid (UltrAu Foil, 600 mesh, R2/1) was cooled to about 100 K in an Elsa cryo-transfer holder (Gatan). While amorphous ice was being deposited on the graphene sheet, the diffraction patterns of the growing ice layer were recorded. During the deposition, the intensity of the molecular diffraction pattern of the amorphous ice initially increased linearly but then leveled off and plateaued once the ice film was so thick that multiple and inelastic scattering became important. From the characteristic curve shape of the diffraction intensity as a function of deposition time, the deposition rate was then determined (typically 0.3 nm s$^{-1}$)[42,50], and the deposition time was adjusted accordingly (typically 74 s to deposit about 10 nm on each side of the sample, 20 nm total).

To perform the deposition and revitrification experiment, the cryo-EM sample was loaded into the microscope with a second Elsa cryo-transfer holder (Gatan, approximately at 100 K). To begin the deposition, the shutter of the specimen holder was opened for the predetermined amount of time, before the shutter was closed to stop any further deposition of amorphous ice. The gas dosing valve was then closed, and the cryo shield surrounding the sample was cooled to liquid nitrogen temperature. Once the pressure in the microscope column had dropped to below $5 \times 10^{-5}$ Pa, the shutter of the cryo-transfer holder was opened, and the revitrification experiment was performed using rectangular laser pulses, as described above.

### Data collection and analysis—T20S proteasome

The T20S datasets were recorded on a Titan Krios G3i, equipped with a BioQuantum energy filter and a K3 camera, at the Center for Microscopy and Image Analysis. The videos were recorded as tiff images in super-resolution mode and binned on the fly to a pixel size of 0.651 Å. The energy filter slit was set to 20 eV. The data were collected using a 100 µm objective aperture at a nominal magnification of ×130,000 during a 1.3-s exposure with a defocus range of −0.6 µm to −2.0 µm and a total dose of 67 e$^-$ Å$^{-2}$.

The datasets were processed in cryoSPARC v4.4 and v4.5 (ref. 5). For the conventional sample, the revitrified sample and the sample revitrified after deposition of amorphous ice, 3,479, 6,105 and 3,641 micrographs were collected, respectively. The micrographs were subjected to patch motion-correction and patch contrast transfer function (CTF) estimation before manual curation on the basis of CTF resolution estimation, ice thickness and total full-frame motion. Particles were initially picked with a blob picker with a diameter of 100–200 Å from 1,519, 2,287 and 2,255 high-quality micrographs for the conventional sample, the revitrified sample and the sample revitrified after deposition of amorphous ice, respectively, and were then extracted with a box size of 512 px and down sampled to 128 px. After two rounds of two-dimensional (2D) classifications, the selected particles were used for ab initio reconstruction with two classes. The particles assigned to the better resolved class were refined against the obtained volume in a homogeneous refinement with $C_1$ symmetry. The resulting volume was used to generate templates for template-based picking with a particle diameter of 150 Å. The particles were again extracted with a box size of 512 px, Fourier-cropped to a box size of 256 px, and subjected to two rounds of 2D classification. The selected particles were used for ab initio reconstruction with only one class, followed by homogeneous refinement with $D_7$ symmetry. Subsequently, 50,000

particles were randomly selected, re-extracted at full box size and again directed into a homogeneous refinement with $D_7$ symmetry. To ensure the comparability of the datasets obtained from the different experimental conditions, a low-pass-filtered volume (filtered to 30 Å) obtained from the conventional dataset was used as a reference input for homogeneous refinements for all the three datasets. Finally, an orientation diagnostics job in cryoSPARC was run to assess the angular distribution of the particles.

## Data collection and analysis—50S ribosomal subunit

The conventional, revitrified and deposited and revitrified 50S datasets were collected on a Titan Krios G4 equipped with a Falcon 4 camera at the Dubochet Center for Imaging facility in Lausanne, Switzerland. The videos were collected in eer format, with a 100 μm objective aperture inserted, at a nominal magnification of ×120,000, corresponding to a pixel size of 0.658 Å per pixel. Micrographs were collected with a defocus range of −0.4 μm to −1 μm using a total dose of 40 e$^-$ Å$^{-2}$. The sample that had been revitrified with the shaped laser pulse and the corresponding conventional control were collected on a Titan Krios G3i, fitted with a BioQuantum energy filter and a K3 camera, at the Center for Microscopy and Image Analysis. The videos were recorded during a 1.2-s exposure at ×130,000 magnification, corresponding to a pixel size of 0.3255 Å in super-resolution mode. The data were binned on the fly to a pixel size of 0.651 Å per pixel. The energy filter slit was set to 20 eV and the objective aperture to 100 μm. The defocus used ranged from −0.4 μm to −1.6 μm, whereas the electron dose was 63 e$^-$ Å$^{-2}$.

For the 50S datasets, 11,121, 6,174, 9,532, 7,416 and 7,725 micrographs were collected for conventional, revitrified, deposited and revitrified, conventional (control for shaped pulses) and shaped pulse revitrified, respectively. The micrographs were patch motion-corrected (40 fractions, no up-sampling), and patch CTF-estimated before being subjected to a manual curation in which low-quality micrographs, judged by CTF estimation, relative ice thickness and total full-frame motion, were discarded. The selected 6,236, 5,870, 6,641, 3,485, and 5,227 high-quality micrographs were subjected to a blob picker with a particle diameter of 200–300 Å. The particles were extracted with a box size of 784 px and cropped to 392 px. After two rounds of 2D classification, selected particles were directed into an ab initio reconstruction with two to three classes. The particles from the best class were refined against the volumes from the best and worst class in one round of heterogeneous refinement. The selected particles from the heterogeneous refinement were directed into a homogeneous refinement with $C_1$ symmetry. Subsequently, 50,000 particles were randomly selected, re-extracted at full box size and refined through homogeneous refinement using $C_1$ symmetry and a low-pass filtered volume from the conventional dataset as a reference. Ultimately, an orientation diagnostics job in cryoSPARC was run to assess the angular distribution of the particles. Unlike the other reconstructions in this manuscript, the 50S ribosomal subunit maps were not sharpened to preserve the details in the flexible L1 stalk.

## Data collection and analysis—HIV-1 envelope ectodomain

The HIV-1 datasets were collected on a Titan Krios G4 using a Falcon 4i camera at the Dubochet Center for Imaging. The videos were recorded in eer format using a magnification of ×96,000, resulting in a pixel size of 0.83 Å per pixel. A total dose of 50 e$^-$ Å$^{-2}$ was used and the defocus range was set to −1 μm to −2.4 μm.

For the conventional HIV-1 dataset, 2,614 micrographs were collected. For the revitrified dataset, data were collected from two different grids; from the first grid, 5,447 micrographs were collected; from the second, 5,081 images were recorded. All datasets were subjected patch motion correction, patch CTF estimation and a manual curation based on CTF resolution estimation, relative ice thickness and the defocus range. Next, the particles were picked using a blob picker with a diameter of 100–150 Å. The particles were extracted with a box

size of 480 px and down sampled to 240 px. The particles from the two revitrified datasets were combined before sorting in two rounds of two-dimensional classification. The selected particles were subjected to an ab initio reconstruction using two classes. Particles were further classified in three dimensions through two rounds of heterogeneous refinements, in which all selected particles from the first round of 2D classification were sorted against the good initial volume and five decoy classes, generated through the pre-mature termination of an ab initio reconstruction. The selected set of particles was then refined in a non-uniform refinement with $C_1$ symmetry. Next, the particles were re-extracted at full box size, locally and globally CTF-refined before being directed into another non-uniform refinement, without dynamic masking and symmetry enforcement. Finally, 50,000 random particles were selected and again refined in a non-uniform refinement without dynamic masking using $C_1$ symmetry and the filtered conventional volume as a reference, before being directed into an orientation diagnostics job.

## Data collection and analysis—hemagglutinin

The four hemagglutinin datasets were recorded on a Titan Krios G4 equipped with a Falcon 4i and a SelectrisX energy filter at the Dubochet Center for Imaging. The videos were collected in eer format at a magnification of ×165,000, corresponding to a pixel size of 0.732 Å. A total dose of 40 e$^-$ Å$^{-2}$ was used for the conventional, revitrified and leading-edge revitrified and deposited and revitrified dataset, respectively. A defocus range of −0.8 μm to −2.5 μm and an energy filter slit of 10 eV was applied for all four datasets.

For the conventional, revitrified, deposited and revitrified and shaped pulse revitrified HA samples, 2,251, 6,476, 10,114 and 10,270 micrographs were collected. The micrographs were subjected to patch motion correction and patch CTF estimation. The micrographs were manually curated on the basis of their CTF resolution estimation, relative ice thickness and total full-frame motion, resulting in 1,651, 4,608, 7,982 and 8,590 high-quality micrographs. On 100 micrographs of each dataset, a denoiser was trained and subsequently applied to the curated micrographs. Particles were picked from denoised micrographs using a blob picker with a diameter of 80–140 Å. Next, the particles were extracted from the raw, non-denoised micrographs using a box size of 360 px, which was cropped to 180 px. The extracted particles were subjected to one to two rounds of 2D classification, followed by an ab initio reconstruction with two classes. The particles were then further cleaned up in one round of heterogeneous refinement using a decoy volume and the best-resolved ab initio volume as inputs. The sorted particles were then refined in a homogeneous refinement, before being re-extracted at full box size. Out of these, 50,000 random particles were selected and then subjected into a final homogeneous refinement with $C_3$ symmetry, using the filtered conventional map as a reference, and an orientation diagnostics job.

## Reporting summary

Further information on research design is available in the Nature Portfolio Reporting Summary linked to this article.

## Data availability

Supplementary Information is available for this paper.
The cryo-EM maps have been deposited in the Electron Microscopy Data Bank (EMDB) and the Electron Microscopy Public Image Archive (EMPIAR) under accession codes EMD-51744 and EMPIAR-12389 (T20S conventional), EMD-51745 and EMPIAR-12388 (T20S revitrified), EMD-51746 and EMPIAR-12390 (T20S revitrified after deposition), EMD-51747 and EMPIAR-12397 (50S conventional), EMD-51748 and EMPIAR-12398 (50S revitrified), EMD-51749 and EMPIAR-12399 (50S revitrified after deposition), EMD-51750 and EMPIAR-12435 (50S shaped pulse conventional), EMD-51751 and EMPIAR-12436 (50S shaped pulse revitrified), EMD-51752 and EMPIAR-12437 (HIV conventional), EMD-51753 and

EMPIAR-12438 (HIV revitrified), EMD-51754 and EMPIAR-12439 (HA conventional), EMD-51755 and EMPIAR-12440 (HA revitrified), EMD-51756 and EMPIAR-12441 (HA revitrified after deposition), and EMD-51757 and EMPIAR-12442 (HA shaped pulse revitrified).

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

## Acknowledgements

We thank B. Beckert (Dubochet Center for Imaging Lausanne), A. Guskov (University of Gröningen) and R. Henderson (Duke University) for providing the protein samples used in this study. A. Myasnikov, B. Beckert, S. Nazarov, I. Mohammed and E. Uchikawa, the members of the Dubochet Center for Imaging in Lausanne, and P. Szwedziak at the Center for Microscopy and Image Analysis in Zurich, are acknowledged for their assistance with cryo-EM data collection. We acknowledge current and former members of the lab for their help at various stages of the project. The work was supported by Swiss National Science Foundation Grant TMCG-2_213773 (U.J.L.) and by the Duke Center for HIV Structural Biology, NIH grant U54AI170752 (U.J.L.).

## Author contributions

U.J.L. was responsible for conceptualizing this work. The methodology was performed by M.S.S., O.F.H., N.J.M., S.V.B. and J.H. M.S.S., O.F.H., N.J.M., S.V.B. and J.H. prepared the cryo-EM samples. M.S.S., S.V.B. and J.H. acquired the cryo-EM data. M.S.S. and S.V.B. processed cryo-EM data. N.J.M. performed the simulations. M.S.S., M.D. and U.J.L. performed the data analysis. U.J.L. acquired funding, handled project administration and supervised the project. The writing of the original draft and data visualization were done by M.S.S. and U.J.L. The reviewing and editing of the paper were performed by M.S.S. and U.J.L. with the input of all coauthors.

## Competing interests

The authors filed for three patents: patent application US-20250052992-A1 "Microsecond melting and revitrification of cryo samples with a correlative light electron microscopy setup"; publication date 13 February 2025; (OFH, MD, UJL). Patent application US 63/555,160 "Method to overcome preferred orientation in cryo-samples for single particle analysis" filed on 19 February 2024 (MSS, OFH, NJM, MD, UJL). Patent application US 63/767,702 "High-resolution liquid cells for microsecond time-resolved cryo-EM"; filed on 6 March 2025 (JH, MD, UJL).

## Additional information

**Correspondence and requests for materials** should be addressed to Ulrich J. Lorenz.

# Reporting Summary

## Statistics

For all statistical analyses, confirm that the following items are present in the figure legend, table legend, main text, or Methods section.

| n/a | Confirmed | |
|---|---|---|
| ☐ | ☒ | The exact sample size (*n*) for each experimental group/condition, given as a discrete number and unit of measurement |
| ☒ | ☐ | A statement on whether measurements were taken from distinct samples or whether the same sample was measured repeatedly |
| ☒ | ☐ | The statistical test(s) used AND whether they are one- or two-sided<br>*Only common tests should be described solely by name; describe more complex techniques in the Methods section.* |
| ☒ | ☐ | A description of all covariates tested |
| ☒ | ☐ | A description of any assumptions or corrections, such as tests of normality and adjustment for multiple comparisons |
| ☐ | ☒ | A full description of the statistical parameters including central tendency (e.g. means) or other basic estimates (e.g. regression coefficient) AND variation (e.g. standard deviation) or associated estimates of uncertainty (e.g. confidence intervals) |
| ☒ | ☐ | For null hypothesis testing, the test statistic (e.g. *F*, *t*, *r*) with confidence intervals, effect sizes, degrees of freedom and *P* value noted<br>*Give P values as exact values whenever suitable.* |
| ☒ | ☐ | For Bayesian analysis, information on the choice of priors and Markov chain Monte Carlo settings |
| ☒ | ☐ | For hierarchical and complex designs, identification of the appropriate level for tests and full reporting of outcomes |
| ☒ | ☐ | Estimates of effect sizes (e.g. Cohen's *d*, Pearson's *r*), indicating how they were calculated |

*Our web collection on statistics for biologists contains articles on many of the points above.*

## Software and code

Policy information about availability of computer code

| Data collection | *Provide a description of all commercial, open source and custom code used to collect the data in this study, specifying the version used OR state that no software was used.* |
|---|---|
| Data analysis | cryoSPARC v.4.4 - v.4.6, ChimeraX v1.7-1.8 |

For manuscripts utilizing custom algorithms or software that are central to the research but not yet described in published literature, software must be made available to editors and reviewers. We strongly encourage code deposition in a community repository (e.g. GitHub). See the Nature Portfolio guidelines for submitting code & software for further information.

## Data

Policy information about availability of data

All manuscripts must include a data availability statement. This statement should provide the following information, where applicable:
- Accession codes, unique identifiers, or web links for publicly available datasets
- A description of any restrictions on data availability
- For clinical datasets or third party data, please ensure that the statement adheres to our policy

The data that support the findings of this study are available from the corresponding author upon request. The cryo-EM maps have been deposited in the Electron Microscopy Data Bank (EMDB) and the Electron Microscopy Public Image Archive (EMPIAR) under accession codes EMD-51744 and EMPIAR-12389 (T20S conventional), EMD-51745 and EMPIAR-12388 (T20S revitrified), EMD-51746 and EMPIAR-12390 (T20S revitrified after deposition), EMD-51747 and EMPIAR-12397

(50S conventional), EMD-51748 and EMPIAR-12398 (50S revitrified), EMD-51749 and EMPIAR-12399 (50S revitrified after deposition), EMD-51750 and EMPIAR-12435 (50S shaped pulse conventional), EMD-51751 and EMPIAR-12436 (50S shaped pulse revitrified), EMD-51752 and EMPIAR-12437 (HIV conventional), EMD-51753 and EMPIAR-12438 (HIV revitrified), EMD-51754 and EMPIAR-12439 (HA conventional), EMD-51755 and EMPIAR-12440 (HA revitrified), EMD-51756 and EMPIAR-12441 (HA revitrified after deposition), and EMD-51757 and EMPIAR-12442 (HA shaped pulse revitrified).

## Human research participants

Policy information about studies involving human research participants and Sex and Gender in Research.

| | |
|---|---|
| Reporting on sex and gender | No such information was collected. |
| Population characteristics | see above |
| Recruitment | No participants were recruited for this study. |
| Ethics oversight | No ethics organization needed to oversee the study. |

Note that full information on the approval of the study protocol must also be provided in the manuscript.

# Field-specific reporting

Please select the one below that is the best fit for your research. If you are not sure, read the appropriate sections before making your selection.

☒ Life sciences      ☐ Behavioural & social sciences      ☐ Ecological, evolutionary & environmental sciences

For a reference copy of the document with all sections, see nature.com/documents/nr-reporting-summary-flat.pdf

# Life sciences study design

All studies must disclose on these points even when the disclosure is negative.

| | |
|---|---|
| Sample size | No sample size calculations were performed. 50'000 random particles were chosen for each reconstruction to have an equal number of particles per reconstruction. This allowed for comparisons of the orientation distribution between the experimental conditions. |
| Data exclusions | Particles , which did not contain any useful information, were excluded during 2D classification and 3D classification through ab initio reconstruction and heterogeneous refinement. The exclusion was performed by assessment of the classification results (whether the classes correspond to the protein of interest or not). |
| Replication | The applicability of the methods was tested on two to four different and independent protein systems. Each protein system was investigated once. |
| Randomization | No randomization was performed, as there were no experimental groups in this study. |
| Blinding | No blinding was performed, as there were no experimental groups in this study. |

# Reporting for specific materials, systems and methods

We require information from authors about some types of materials, experimental systems and methods used in many studies. Here, indicate whether each material, system or method listed is relevant to your study. If you are not sure if a list item applies to your research, read the appropriate section before selecting a response.

### Materials & experimental systems

| n/a | Involved in the study |
|---|---|
| ☒ ☐ | Antibodies |
| ☒ ☐ | Eukaryotic cell lines |
| ☒ ☐ | Palaeontology and archaeology |
| ☒ ☐ | Animals and other organisms |
| ☒ ☐ | Clinical data |
| ☒ ☐ | Dual use research of concern |

### Methods

| n/a | Involved in the study |
|---|---|
| ☒ ☐ | ChIP-seq |
| ☒ ☐ | Flow cytometry |
| ☒ ☐ | MRI-based neuroimaging |

