## [Peer Review File · Nature Methods]

Laser Flash Melting Cryo-EM Samples to Overcome Preferred Orientation

Corresponding Author: Dr Ulrich Lorenz

Version 0:

Decision Letter:

31st Jan 2025

Dear Ulrich,

Your Article, "Laser Flash Melting Cryo-EM Samples to Overcome Preferred Orientation", has now been seen by 3 reviewers. As you will see from their comments below, although the reviewers find your work of considerable potential interest, they have raised several concerns. We are interested in the possibility of publishing your paper in Nature Methods, but would like to consider your response to these concerns before we reach a final decision on publication.

We therefore invite you to revise your manuscript to address these concerns. In particular, we think that Reviewer #1's comments about applying the method to an additional small protein and carefully assessing possible sample damage will be important to address.

Link Redacted

We hope to receive your revised paper within 8 weeks. If you cannot send it within this time, please let us know. In this event, we will still be happy to reconsider your paper at a later date so long as nothing similar has been accepted for publication at Nature Methods or published elsewhere.

OPEN SCIENCE REQUIREMENTS

REPORTING SUMMARY AND EDITORIAL POLICY CHECKLISTS

EXTENDED DATA FIGURES

DATA AVAILABILITY

All novel DNA and RNA sequencing data, protein sequences, genetic polymorphisms, linked genotype and phenotype data, gene expression data, macromolecular structures, and proteomics data must be deposited in a publicly accessible database, and accession codes and associated hyperlinks must be provided in the "Data Availability" section.

MATERIALS AVAILABILITY

SUPPLEMENTARY PROTOCOL

To help facilitate reproducibility and uptake of your method, we ask you to prepare a step-by-step Supplementary Protocol for the method described in this paper. We [encourage authors to share their step-by-step experimental protocols](https://www.nature.com/nature-research/editorial-policies/reporting-standards#protocols) on a protocol sharing platform of their choice and report the protocol DOI in the reference list. Nature Portfolio's protocols.io is a free-to-use and open resource for protocols; protocols deposited onto protocols.io are citable and can be linked from the published article. More details can found at [protocols.io](https://www.protocols.io/help/publish-articles).

ORCID

Sincerely yours,
Allison

Allison Doerr, Ph.D.
Chief Editor
Nature Methods

Reviewers' Comments:

Reviewer #1 (Remarks to the Author):

Straub, Harder and Mowry et al describe a highly novel method for overcoming the preferred orientation problem for cryo-EM specimen preparation. The manuscript builds on prior work describing temporary rapid melting and re-vitrification of specimens using defined laser pulses for a highly symmetric and large molecular weight specimen. The authors extend that premise in the current study by utilizing four specimens that attempt to recapitulate the broad utility of this system as a generalizable solution for the preferred orientation problem. The authors vary laser pulse sequence parameters to assess whether that can scramble the orientation of embedded particles. The data presented is well organized, methodical and exhaustive to draw suitable conclusions. However, although the authors do show significant changes in the preferred orientation of some specimens, to this reviewer the utility of this technique is not yet broadly generalizable or substantially better than existing approaches. Currently, detergent optimization, substrate support films and specimen stage tilt are two strategies to overcome preferred orientation. Detergent optimization and use of substrate support films, as the authors note, are laborious and not readily generalizable for all specimens. While specimen stage tilt is generalizable, it is not well suited for specimens smaller than ~300-400 kDa with an obvious impact on global resolution. Sample revitrification can therefore be most useful as a generalizable strategy for smaller specimens. The authors must minimally show that for another specimen where the size is small (less than ~300 kDa) and the SCF* is substantially less than 0.81, the revitrification strategy can improve orientation distribution resulting in SCF* approaching 0.81 or above.

Major comments-

1. Although the technique does not appear to substantially damage any of the four tested specimens, the authors should comment on its broad utility and potential for specimen damage upon revitrification. To this reviewer that is a genuine possibility precluding its widespread use. For example, in the case of 20S proteasome, the local resolution varies substantially between conventional vitrification and laser pulse melted and revitrified specimen in some locations (compare S1f and S2f). Authors should provide zoomed in images of EM density in those locations to clarify how those densities change and whether it is due to increase in specimen damage. Although by this same measure the local resolution improves for 50S ribosome (compare S4f and S5f), which is presumably due to better sampling of missing views, it would be good to assess changes between zoomed in EM densities of these regions of differences between maps. Additionally, the L1 stalk region of 50S ribosome visually looks substantially different upon revitrification (Fig 1d, 2c, S6), while deposition of amorphous ice appears to reduce this effect in the L1 stalk region (Fig 3d). Authors should provide comparative zoomed in images for that region to assess map quality and to rule out substantial local specimen damage.
2. In the same vein, not all samples with preferred orientation tested in this study actually benefit substantially from this method with sampling compensation factor not showing substantial improvements. For 20S proteasome and HIV envelope ectodomain, the SCF indicates already well sampled orientation distributions. For HA trimer the revitrification strategy does not help in suitable improvements in orientation distributions. Although 50S ribosome substantially improves in orientation distribution, it is not a widely generalizable example.

Minor comments-

1. Several of the image panels in the Supplementary section are incomplete like S1a, S1b, S2a, S2b, S3a, S3b, S4b, S5a, S5b, S7a, S7b, S8a, S9a, S10a, S11a, S12a, S13a, S14a and S15a.
2. Several image panels appear pixelated like most of the 'e' and 'g' panels of supplementary figures. Authors should attempt to provide images with better resolution.
3. "The SCF* takes values between 0 and 1, with 1 corresponding to a perfectly isotropic angular distribution, and SCF* values below 0.81 indicating strong preferred orientation." For this statement, authors should amend to indicate that values approaching 0.81 may still be fine to derive a suitable map, however, values progressively lower than 0.81 will have more severe orientation bias. A value of 0.81 indicates perfect side like views, which is sufficient to derive a completely sampled map in Fourier space.

Reviewer #2 (Remarks to the Author):

Preferential particle orientation is a significant problem in single-particle cryo-electron microscopy (cryo-EM), ranging from severe cases that prevent successful structure determination to less severe cases that still negatively affect the final 3D map quality. The authors present convincing evidence that rapid laser melting and re vitrification significantly improves the problem of preferred particle orientation.

The authors selected several test cases and the method worked on all of them to varying degrees. The 50S ribosomal subunit was chosen as a challenging and appropriate test case for this problem, and it resulted in not only improved particle orientation but also a significant improvement in resolution.

It's important to note that improvements in particle orientation do not always translate into higher resolution, as the Fourier Shell Correlation (FSC) resolution criterion is not very sensitive to these issues. However, reducing the preferred orientation will always improve reconstruction quality.

Laser melting combined with an extra layer of water is an excellent idea to address this problem and can give even better results than laser melting alone. Although it requires a new hardware setup in cryo-EM labs, it offers significant advantages: it can be routinely applied to virtually any macromolecular complex, it consistently improves the quality and sometimes the resolution of the calculated 3D maps, and it works independently of biochemical problems or technically challenging tilting experiments. In conclusion, this new laser melting technique represents a promising solution to the long-standing problem of preferential particle orientation in cryo-EM, and offers potential improvements in cryo-EM for a wide range of macromolecular complexes.

This is a highly innovative, broadly applicable and interesting approach and I very much support the publication in Nature Methods.

Reviewer #3 (Remarks to the Author):

Review for Nature Methods Manuscript

The manuscript entitled "Laser Flash Melting Cryo-EM Samples to Overcome Preferred Orientation" by Straub et al., with corresponding author Ulrich J. Lorenz, presents an innovative and highly practical solution to the persistent issue of preferred orientation in cryo-electron microscopy (cryo-EM) sample preparation. The authors introduce an approach involving laser flash melting and re vitrification, which effectively disrupts protein adsorption at the air-water interface. This method allows for significant randomization of protein orientations and improves angular distributions for a range of protein sizes and symmetries, addressing a major bottleneck in achieving high-resolution reconstructions. The manuscript is well written, novel, appropriately referenced, and the data well presented.

Key Strengths of the Study

Innovative Methodology:

The use of laser flash melting to overcome preferred orientation represents a creative and impactful advancement. The authors successfully demonstrate that the method can be optimized by adjusting parameters such as heating rates or by depositing amorphous ice layers prior to re vitrification. This highlights the method's flexibility and adaptability.

Mechanistic Insights:

The manuscript provides valuable insights into the competing processes at play during laser re vitrification. The detailed explanation of particle detachment and diffusion back to the interface adds depth to the study and suggests further avenues for experimental optimization.

Significant Improvements:

The reported 1.2 Å resolution improvement for the 50S ribosomal subunit is a compelling demonstration of the method's potential. Such an improvement, achieved under otherwise identical conditions, highlights the significant reduction in time and resources required for data collection.

Ease of Integration:

One of the most notable aspects of this method is its compatibility with existing workflows. The approach does not require substantial modifications to sample preparation protocols and is feasible for integration into both current cryo-EM systems and correlative light-electron microscopy setups. Additionally, the potential for on-the-fly implementation during data acquisition could make this method even more accessible to cryo-EM facilities.

Broad Applicability:

The manuscript demonstrates the technique's versatility across proteins of varying sizes and symmetries, suggesting its utility across a wide range of cryo-EM applications.

Questions and Recommendations for the Authors

Adaptation to Next-Generation Devices:

Would the authors consider how this methodology could be implemented in the new generation of vitrification devices currently on the market? Could they propose a framework or experimental setup for such integration?

Impact on High-Resolution Data:

For resolutions between 1.3 and 2.0 Å, could the authors provide a more rigorous discussion on the potential limitations of their method? It would be valuable to address whether the laser revitrification process might introduce artifacts or negatively affect very high-resolution reconstructions. A clear presentation of the advantages and potential drawbacks of the method in this resolution range would strengthen the manuscript.

Future Directions:

Could the authors elaborate on how the proposed modifications to interfacial properties—such as vapor deposition of hydrophilic compounds—might be systematically explored in future studies? Would vitrifying in a hydrophilic gas be beneficial?

Overall Assessment

This study is a significant contribution to the cryo-EM field and has the potential to revolutionize sample preparation workflows. The authors provide a well-rounded discussion, compelling experimental evidence, and actionable insights that address a critical challenge in structural biology.

I recommend this manuscript for publication in Nature Methods provided the authors address the remaining questions satisfactorily. These clarifications will further solidify the impact and applicability of this innovative methodology.

Version 1:

Decision Letter:

Our ref: NMETH-A58771A

25th Apr 2025

Dear Ulrich,

Thank you for submitting your revised manuscript "Laser Flash Melting Cryo-EM Samples to Overcome Preferred Orientation" (NMETH-A58771A). It has now been seen by the original referees and their comments are below. The reviewers find that the paper has improved in revision, and therefore we'll be happy in principle to publish it in Nature Methods, pending minor revisions to satisfy the referees' final requests and to comply with our editorial and formatting guidelines.

TRANSPARENT PEER REVIEW

Please note: we allow redactions to authors' rebuttal and reviewer comments in the interest of confidentiality. If you are concerned about the release of confidential data, please let us know specifically what information you would like to have removed. Please note that we cannot incorporate redactions for any other reasons. Reviewer names will be published in the peer review files if the reviewer signed the comments to authors, or if reviewers explicitly agree to release their name. For more information, please refer to our [FAQ](https://www.nature.com/documents/nr-transparent-peer-review.pdf).

ORCID

Sincerely yours,
Allison

Allison Doerr, Ph.D.
Chief Editor
Nature Methods

Reviewer #1 (Remarks to the Author):

The authors have addressed all my minor comments, however, responses to a few major comments require clarification. It is recommended that the authors include the caveats (described below) in the conclusion/discussion section.

“We are not optimistic that data for another sub-300 kDa protein would provide significant additional insights. Our experiments suggest that small proteins differ from larger ones in that they have shorter diffusion times, so that they can reach the interface more rapidly and sample different orientations. This seems to allow Hemagglutinin to find its preferred orientation efficiently, which makes it more challenging to scramble its orientation. In contrast, we do obtain a reduction in preferred orientation for the HIV-1 Env ectodomain protein. An experiment on an additional small protein would likely yield a result somewhere between these two cases. We do not feel that this diminishes the novelty of our approach or its practical utility for a wide range of proteins.”

The author's argument that another sub-300 kDa may not add to the novelty of the approach is perfectly acceptable, however, it does impact the technique's practical utility for smaller proteins. The scrambling of orientations for HIV-1 Env ectodomain to improve SCF from 0.89 to 0.99 indicates improvement of an already well sampled specimen in Fourier space. Directional resolution for this specimen is almost identical for the 0.89 and 0.99 SCF maps (see supplementary fig 9e and 10e). An improvement in SCF only for pathologically preferred orientations is of substantial benefit for final map quality. The authors should include this caveat in their discussion/conclusion that their method will likely face challenges for progressively smaller proteins owing to the explanation provided by the authors that diffusion to the air-water interface is a lot quicker for smaller proteins.

“With the exception of Hemagglutinin, the SCF* value increases upon flash melting and revitrification for all examples shown. For the HIV-1 Env ectodomain protein, flash melting even results in a near-perfect uniform distribution (SCF* = 0.99). A more even particle distribution stands to improve the quality of the details of the map. Another way to look at this is that it has now become more common to “rebalance” the particle distribution during reconstruction or for the characterization of conformational ensembles. This however usually amounts to rejecting large numbers of particles, which is costly. By reshuffling the particle orientations in the sample, this can be avoided.”

Like my response above, the utility of rebalancing particles is more pronounced and beneficial to map quality only if the specimen has pathologically preferred orientation bias to begin with (SCF substantially lower than 0.81). Even in the example provided, post revitrification the map with an SCF of 0.99 has a drop in resolution by ~0.6 Å compared to the map with an SCF of 0.89 and in supplementary fig 10e (bottom right) the revitrified map appears to have lost the resolution of an alpha helix relative to the conventional map. Such a drop with an identical particle number in the particle stack would imply the method still faces challenges for smaller specimens.

Reviewer #3 (Remarks to the Author):

Reply is perfect!

Version 2:

Decision Letter:

16th Jul 2025

Dear Ulrich,

I am pleased to inform you that your Article, "Laser Flash Melting Cryo-EM Samples to Overcome Preferred Orientation", has now been accepted for publication in Nature Methods. The received and accepted dates will be 21 November 2024 and 16 July 2025. This note is intended to let you know what to expect from us over the next month or so, and to let you know where to address any further questions.

Over the next few weeks, your paper will be copyedited to ensure that it conforms to Nature Methods style. Once your paper is typeset, you will receive an email with a link to choose the appropriate publishing options for your paper and our Author Services team will be in touch regarding any additional information that may be required. It is extremely important that you let us know now whether you will be difficult to contact over the next month. If this is the case, we ask that you send us the contact information (email, phone and fax) of someone who will be able to check the proofs and deal with any last-minute problems.

Authors may need to take specific actions to achieve compliance with funder and institutional open access mandates. If your research is supported by a funder that requires immediate open access (e.g. according to [a href="https://www.springernature.com/gp/open-science/plan-s-compliance"> Plan S principles](https://www.springernature.com/gp/open-science/plan-s-compliance) or the a

<https://www.springernature.com/gp/open-science/us-federal-agency-compliance>) then you should select the gold OA route, and we will direct you to the compliant route where possible. Because authors warrant under our subscription licensing terms that they haven't committed to licensing any version of their article under a licence inconsistent with the terms of our agreement – including the applicable embargo period – publication under the subscription model isn't suitable for authors whose funders require no embargo.

Best regards,
Allison

Allison Doerr, Ph.D.
Chief Editor
Nature Methods

** Visit the Springer Nature Editorial and Publishing website at http://editorial-jobs.springernature.com?utm_source=ejP_NMeth_email&utm_medium=ejP_NMeth_email&utm_campaign=ejp_Nmeth > www.springernature.com/editorial-and-publishing-jobs for more information about our career opportunities. If you have any questions please click [here](mailto:editorial.publishing.jobs@springernature.com) . **

We would like to thank the reviewers for their careful reviews. We are attaching our responses below.

Reviewer #1

Straub, Harder and Mowry et al describe a highly novel method for overcoming the preferred orientation problem for cryo-EM specimen preparation. The manuscript builds on prior work describing temporary rapid melting and re-vitrification of specimens using defined laser pulses for a highly symmetric and large molecular weight specimen. The authors extend that premise in the current study by utilizing four specimens that attempt to recapitulate the broad utility of this system as a generalizable solution for the preferred orientation problem. The authors vary laser pulse sequence parameters to assess whether that can scramble the orientation of embedded particles. The data presented is well organized, methodical and exhaustive to draw suitable conclusions. However, although the authors do show significant changes in the preferred orientation of some specimens, to this reviewer the utility of this technique is not yet broadly generalizable or substantially better than existing approaches.

Currently, detergent optimization, substrate support films and specimen stage tilt are two strategies to overcome preferred orientation. Detergent optimization and use of substrate support films, as the authors note, are laborious and not readily generalizable for all specimens. While specimen stage tilt is generalizable, it is not well suited for specimens smaller than ~300-400 kDa with an obvious impact on global resolution. Sample revitrification can therefore be most useful as a generalizable strategy for smaller specimens. The authors must minimally show that for another specimen where the size is small (less than ~300 kDa) and the SCF* is substantially less than 0.81, the revitrification strategy can improve orientation distribution resulting in SCF* approaching 0.81 or above.

We are not optimistic that data for another sub-300 kDa protein would provide significant additional insights. Our experiments suggest that small proteins differ from larger ones in that they have shorter diffusion times, so that they can reach the interface more rapidly and sample different orientations. This seems to allow Hemagglutinin to find its preferred orientation efficiently, which makes it more challenging to scramble its orientation. In contrast, we do obtain a reduction in preferred orientation for the HIV-1 Env ectodomain protein. An experiment on an additional small protein would likely yield a result somewhere between these two cases.

We do not feel that this diminishes the novelty of our approach or its practical utility for a wide range of proteins.

Major comments-

1. Although the technique does not appear to substantially damage any of the four tested specimens, the authors should comment on its broad utility and potential for specimen damage upon revitrification. To this reviewer that is a genuine possibility precluding its widespread use. For example, in the case of 20S proteasome, the local resolution varies substantially between conventional vitrification and laser pulse melted and revitrified specimen in some locations (compare S1f and S2f). Authors should provide zoomed in images of EM density in those locations to clarify how those densities change and whether it is due to increase in specimen damage. Although by this same measure the local resolution improves for 50S ribosome (compare S4f and S5f), which is presumably due to better sampling of missing views, it would be good to assess changes between zoomed in EM densities of these regions of differences between maps. Additionally, the L1 stalk region of 50S ribosome visually looks substantially different upon revitrification (Fig 1d, 2c, S6), while deposition of amorphous ice appears to reduce this effect in the L1 stalk region (Fig 3d). Authors should provide comparative zoomed in images for that region to assess map quality and to rule out substantial local specimen damage.

Our experiments do not provide any evidence that the particles are damaged by the revitrification process, which is consistent with our previous results, where we have obtained high-resolution reconstructions from revitrified samples (*Acta Crystallogr. D* **79**, 473–478 (2023), *Front. Mol. Biosci.* **9**, 1044509 (2022)). For the T20S proteasome, we have added panels to Figs. S2 and S3 that show a comparison between the reconstructions from a revitrified and conventional cryo-EM sample for the regions in question. This comparison does not provide any evidence of structural damage, but shows that the structures are identical within the resolution obtained. For the 50S ribosomal subunit and the HIV-1 Env ectodomain protein, a similar comparison leads to the same conclusion. We have added corresponding panels to the SI.

As the reviewer points out, the flexible L1 stalk of the 50S ribosomal subunit appears to lose density in the reconstructions of Figs. 1d and 2c. This appears to be an artefact induced by the auto-sharpen tool in cryoSPARC. As the overall resolution improves in revitrified samples thanks to the reduction in preferred orientation, auto-sharpening reduces the density in the less well resolved, flexible regions, such as the L1 stalk. This may create the impression that density is missing, which is not the case. We have therefore decided to show unsharpened maps for the 50S ribosomal subunit throughout the manuscript. These maps make it clear that the particles are not damaged.

2. In the same vein, not all samples with preferred orientation tested in this study actually benefit substantially from this method with sampling compensation factor not showing substantial improvements. For 20S proteasome and HIV envelope ectodomain, the SCF indicates already well sampled orientation distributions. For HA trimer the revitrification strategy does not help in suitable improvements in orientation distributions. Although 50S ribosome substantially improves in orientation distribution, it is not a widely generalizable example.

With the exception of Hemagglutinin, the SCF* value increases upon flash melting and revitrification for all examples shown. For the HIV-1 Env ectodomain protein, flash melting even results in a near-perfect uniform distribution (SCF* = 0.99). A more even particle distribution stands to improve the quality of the details of the map.

Another way to look at this is that it has now become more common to “rebalance” the particle distribution during reconstruction or for the characterization of conformational ensembles. This however usually amounts to rejecting large numbers of particles, which is costly. By reshuffling the particle orientations in the sample, this can be avoided.

Minor comments-

1. Several of the image panels in the Supplementary section are incomplete like S1a, S1b, S2a, S2b, S3a, S3b, S4b, S5a, S5b, S7a, S7b, S8a, S9a, S10a, S11a, S12a, S13a, S14a and S15a.

2. Several image panels appear pixelated like most of the ‘e’ and ‘g’ panels of supplementary figures. Authors should attempt to provide images with better resolution.

Points 1 and 2 appear to be an issue that occurs for some pdf readers. We have tried to fix these compatibility problems.

3. “The SCF* takes values between 0 and 1, with 1 corresponding to a perfectly isotropic angular distribution, and SCF* values below 0.81 indicating strong preferred orientation.” For this statement, authors should amend to indicate that values approaching 0.81 may still be fine to derive a suitable map, however, values progressively lower than 0.81 will have more severe orientation bias. A value of 0.81 indicates perfect side like views, which is sufficient to derive a completely sampled map in Fourier space.

We have simplified the statement. To avoid confusion, we have left out the reference to a distribution with perfectly sampled equatorial views and a corresponding SCF* value of 0.81.

Reviewer #2

Preferential particle orientation is a significant problem in single-particle cryo-electron microscopy (cryo-EM), ranging from severe cases that prevent successful structure determination to less severe cases that still negatively affect the final 3D map quality. The authors present convincing evidence that rapid laser melting and revitrification significantly improves the problem of preferred particle orientation.

The authors selected several test cases and the method worked on all of them to varying degrees. The 50S ribosomal subunit was chosen as a challenging and appropriate test case for this problem, and it resulted in not only improved particle orientation but also a significant improvement in resolution.

It's important to note that improvements in particle orientation do not always translate into higher resolution, as the Fourier Shell Correlation (FSC) resolution criterion is not very sensitive to these issues. However, reducing the preferred orientation will always improve reconstruction quality.

Laser melting combined with an extra layer of water is an excellent idea to address this problem and can give even better results than laser melting alone. Although it requires a new hardware setup in cryo-EM labs, it offers significant advantages: it can be routinely applied to virtually any macromolecular complex, it consistently improves the quality and sometimes the resolution of the calculated 3D maps, and it works independently of biochemical problems or technically challenging tilting experiments. In conclusion, this new laser melting technique represents a promising solution to the long-standing problem of preferential particle orientation in cryo-EM, and offers potential improvements in cryo-EM for a wide range of macromolecular complexes.

This is a highly innovative, broadly applicable and interesting approach and I very much support the publication in Nature Methods.

We would like to thank the reviewer for their comments.

Reviewer #3

Review for Nature Methods Manuscript

The manuscript entitled "Laser Flash Melting Cryo-EM Samples to Overcome Preferred Orientation" by Straub et al., with corresponding author Ulrich J. Lorenz, presents an innovative and highly practical solution to the persistent issue of preferred orientation in cryo-electron microscopy (cryo-EM) sample preparation. The authors introduce an approach involving laser flash melting and revitrification, which effectively disrupts protein adsorption at the air-water interface. This method allows for significant randomization of protein orientations and improves angular distributions for a range of protein sizes and symmetries, addressing a major bottleneck in achieving high-resolution reconstructions. The manuscript is well written, novel, appropriately referenced, and the data well presented.

Key Strengths of the Study

Innovative Methodology:

The use of laser flash melting to overcome preferred orientation represents a creative and impactful advancement. The authors successfully demonstrate that the method can be optimized by adjusting parameters such as heating rates or by depositing amorphous ice layers prior to revitrification. This highlights the method's flexibility and adaptability.

Mechanistic Insights:

The manuscript provides valuable insights into the competing processes at play during laser revitrification. The detailed explanation of particle detachment and diffusion back to the interface adds depth to the study and suggests further avenues for experimental optimization.

Significant Improvements:

The reported 1.2 Å resolution improvement for the 50S ribosomal subunit is a compelling demonstration of the method's potential. Such an improvement, achieved under otherwise identical conditions, highlights the significant reduction in time and resources required for data collection.

Ease of Integration:

One of the most notable aspects of this method is its compatibility with existing workflows. The approach does not require substantial modifications to sample preparation protocols and is feasible for integration into both current cryo-EM systems and correlative light-electron microscopy setups. Additionally, the potential for on-the-fly implementation during data acquisition could make this method even more accessible to cryo-EM facilities.

Broad Applicability:

The manuscript demonstrates the technique's versatility across proteins of varying sizes and symmetries, suggesting its utility across a wide range of cryo-EM applications.

Questions and Recommendations for the Authors

Adaptation to Next-Generation Devices:

Would the authors consider how this methodology could be implemented in the new generation of vitrification devices currently on the market? Could they propose a framework or experimental setup for such integration?

Some vitrification devices that are already commercially available or currently under development in different labs incorporate an optical microscope to assess the sample quality and thickness. Such an optical microscope could be used to perform the melting and revitrification experiments, as we have previously described (*Front. Mol. Biosci.* **9**, 1044509 (2022)). We have added a corresponding remark to the manuscript.

Impact on High-Resolution Data:

For resolutions between 1.3 and 2.0 Å, could the authors provide a more rigorous discussion on the potential limitations of their method? It would be valuable to address whether the laser revitrification process might introduce artifacts or negatively affect very high-resolution reconstructions. A clear presentation of the advantages and potential drawbacks of the method in this resolution range would strengthen the manuscript.

We have previously shown that high-resolution reconstructions of apoferritin can be obtained from revitrified cryo-EM samples, with the resolution comparable to that afforded by conventional cryo-EM samples (*Acta Crystallogr. D* **79**, 473–478 (2023), *Front. Mol. Biosci.* **9**, 1044509 (2022)). This suggests that the revitrification process does not impose a fundamental limit on the spatial resolution that can be obtained. In the work presented here, we show moreover that reshuffling the particle orientations can even be used to improve the quality of the details of the reconstructions. More work is currently underway to characterize the effects of revitrification on high-resolution reconstructions. We have added a corresponding remark to the introduction.

Future Directions:

Could the authors elaborate on how the proposed modifications to interfacial properties—such as vapor deposition of hydrophilic compounds—might be systematically explored in future studies? Would vitrifying in a hydrophilic gas be beneficial?

We are currently finishing a manuscript in which we are exploring this idea and which we hope to submit soon.

Overall Assessment

This study is a significant contribution to the cryo-EM field and has the potential to revolutionize sample preparation workflows. The authors provide a well-rounded discussion, compelling experimental evidence, and actionable insights that address a critical challenge in structural biology.

I recommend this manuscript for publication in *Nature Methods* provided the authors address the remaining questions satisfactorily. These clarifications will further solidify the impact and applicability of this innovative methodology.

We would like to thank the reviewer for their comments.

We would like to thank the reviewers for giving the revised manuscript their thorough consideration.

Reviewer #1

The authors have addressed all my minor comments, however, responses to a few major comments require clarification. It is recommended that the authors include the caveats (described below) in the conclusion/discussion section.

“We are not optimistic that data for another sub-300 kDa protein would provide significant additional insights. Our experiments suggest that small proteins differ from larger ones in that they have shorter diffusion times, so that they can reach the interface more rapidly and sample different orientations. This seems to allow Hemagglutinin to find its preferred orientation efficiently, which makes it more challenging to scramble its orientation. In contrast, we do obtain a reduction in preferred orientation for the HIV-1 Env ectodomain protein. An experiment on an additional small protein would likely yield a result somewhere between these two cases. We do not feel that this diminishes the novelty of our approach or its practical utility for a wide range of proteins.”

The author's argument that another sub-300 kDa may not add to the novelty of the approach is perfectly acceptable, however, it does impact the technique's practical utility for smaller proteins. The scrambling of orientations for HIV-1 Env ectodomain to improve SCF from 0.89 to 0.99 indicates improvement of an already well sampled specimen in Fourier space. Directional resolution for this specimen is almost identical for the 0.89 and 0.99 SCF maps (see supplementary fig 9e and 10e). An improvement in SCF only for pathologically preferred orientations is of substantial benefit for final map quality. The authors should include this caveat in their discussion/conclusion that their method will likely face challenges for progressively smaller proteins owing to the explanation provided by the authors that diffusion to the air-water interface is a lot quicker for smaller proteins.

“With the exception of Hemagglutinin, the SCF* value increases upon flash melting and revitrification for all examples shown. For the HIV-1 Env ectodomain protein, flash melting even results in a near-perfect uniform distribution (SCF* = 0.99). A more even particle distribution stands to improve the quality of the details of the map. Another way to look at this is that it has now become more common to “rebalance” the particle distribution during reconstruction or for the characterization of conformational ensembles. This however usually amounts to rejecting large numbers of particles, which is costly. By reshuffling the particle orientations in the sample, this can be avoided.”

Like my response above, the utility of rebalancing particles is more pronounced and beneficial to map quality only if the specimen has pathologically preferred orientation bias to begin with (SCF substantially lower than 0.81). Even in the example provided, post revitrification the map with an SCF of 0.99 has a drop in resolution by ~0.6 Å compared to the map with an SCF of 0.89 and in supplementary fig 10e (bottom right) the revitrified map appears to have lost the resolution of an alpha helix relative to the conventional map. Such a drop with an identical particle number in the particle stack would imply the method still faces challenges for smaller specimens.

We have edited the conclusion of the manuscript to emphasize that the shorter diffusion times of small proteins make it more challenging to scramble their orientations. As we have stated before, a potential strategy would be to outrun diffusion back to the interface by performing flash melting experiments with shorter laser pulses.

Reviewer #3

Reply is perfect!